# An Evaluation of Ethograms Measuring Distinct Features of Enrichment Use by Captive Chimpanzees (*Pan troglodytes*)

**DOI:** 10.3390/ani12162029

**Published:** 2022-08-10

**Authors:** Julia L. Greeson, Kara I. Gabriel, J. B. Mulcahy, Bonnie King Hendrickson, Susan D. Lonborg, Jay C. Holloway

**Affiliations:** 1Primate Behavior Master’s Program, Central Washington University, Ellensburg, WA 98926, USA; 2Department of Psychology, Central Washington University, Ellensburg, WA 98926, USA; 3Faculty of Primate Behavior & Ecology Program, Central Washington University, Ellensburg, WA 98926, USA; 4Chimpanzee Sanctuary Northwest, Cle Elum, WA 98922, USA; 5Psychology Program, Central Washington University, Ellensburg, WA 98926, USA

**Keywords:** primate behavior, environmental enrichment, *Pan troglodytes*, social contexts, object manipulation, individual preferences, principal component analysis, chimpanzee, primate welfare

## Abstract

**Simple Summary:**

Environmental enrichment for chimpanzees is important in order to minimize boredom and stress in captivity and to provide opportunities to engage in species-typical behaviors. However, few studies have investigated potential associations between enrichment objects, manipulation behaviors, and social contexts, nor have they examined if individual chimpanzees vary in their enrichment object preferences. In the current study, three ethograms were used to code the use of enrichment objects, engagement in manipulation behaviors, and social contexts of enrichment use of captive chimpanzees (*Pan troglodytes*). Data mining from a video archive consisting of 2054 videos collected over a decade allowed the ethograms to be applied to 732 min and 58 s of videos. Some enrichment objects were more often associated with specific manipulation behaviors and social contexts, indicating that enrichment objects might serve distinct social and behavioral purposes. The chimpanzees differed in their enrichment object preferences, suggesting that caregivers of captive chimpanzees should consider individual needs when providing enrichment in order to improve chimpanzees’ experiences in captivity. Finally, the majority of enrichment object use and manipulation behaviors were able to be categorized, indicating that our ethograms were largely effective in coding enrichment use.

**Abstract:**

Environmental enrichment provides mental stimulation and minimizes abnormal behaviors in captive animals. In captive chimpanzees, individual animals may vary in the ways in which they benefit from enrichment or use enrichment devices, so investigating nuances in enrichment use may improve the welfare of captive chimpanzees. In the current study, three ethograms measuring distinct features of enrichment use (i.e., enrichment object, manipulation behavior, and social context) were evaluated by coding videos of captive chimpanzees (*Pan troglodytes*) at Chimpanzee Sanctuary Northwest in Cle Elum, WA. A total of 732 min and 58 s of video footage was coded from a larger video archive (i.e., 2054 videos) of enrichment use that spanned a decade. A principal component analysis (PCA) revealed that different categories of enrichment objects were more often associated with specific manipulation behaviors and social contexts, suggesting that enrichment objects might fulfill different behavioral and social needs in captivity. Specifically, toy objects were associated with active tactile behaviors in affiliative contexts while oral behaviors were used with foraging objects in solitary contexts. Additionally, individual chimpanzees showed unique preferences for enrichment objects, indicating that caregivers of captive chimpanzees should consider individual needs instead of a “one size fits all” approach to enrichment provisions.

## 1. Introduction

Many chimpanzees are housed in captive settings that are largely different from their natural habitats. Furthermore, captive chimpanzees are prone to experience boredom and stress, which can potentially manifest as abnormal behaviors such as self-harm or repetitive behaviors that do not overtly appear to serve any purpose [1,2,3]. Primates, in particular, benefit from environmental enrichment which can be used to improve physical and psychological wellbeing, minimize stress and boredom, reduce engagement in abnormal behaviors, and promote the expression of natural behaviors [4,5]. The use of enrichment devices in primate care has become a commonly accepted professional practice. In a 2014 survey of facilities housing laboratory nonhuman primates, 100% of those facilities reported supplying task-oriented items that provided extractive, manipulative, or cognitive challenges to caged primates [6].

The United States Department of Agriculture Animal Care Blue Book defines environmental enrichment as the “means of expressing noninjurious species-typical activities” [7] (p. 175), noting that species differences should be considered. Complexity, predictability, controllability, and novelty can all influence the impact of enrichment devices on animal welfare; enrichment objects that are biologically relevant, account for an animal’s behavioral ecology, and elicit natural behaviors may be particularly effective [4,5]. To ensure that enrichment objects are improving an animal’s welfare, researchers and animal caregivers often observe and compare primates’ activity levels and behaviors before and after objects are provided (e.g., [8,9,10,11]). For example, artificial termite-fishing tasks decreased inactivity and increased tool use in captive chimpanzees [11].

Observing enrichment use may also help identify those objects that animals are more motivated to use or that provide the most benefits. For example, when chimpanzees were provided televisions, balls, and mirrors, they interacted with televisions the most and mirrors the least [12]. In a comparison of nesting materials, affiliative behaviors increased and abnormal behaviors decreased in the presence of any nesting materials even though individual chimpanzees favored shredded newspaper over other materials such as leaves and branches, long grass, and cotton sheets [13]. In another example, using a painting application on a digital device was as effective as painting with a brush on paper at reducing some but not all stereotypic and displacement behaviors in zoo-housed chimpanzees [14].

Examining group preferences for objects can help caregivers ensure that the majority of captive chimpanzees are provided with appealing devices. However, differences in enrichment needs and preferences in nonhuman primates may emerge at the individual level according to age, sex, and other behavioral and environmental variables [15]. Differences in enrichment use by chimpanzees have been shown based on age [8,12,16,17,18,19], dominance rank [20], sex [21,22], housing conditions [16,19], and degree of baseline stereotypies [8]. Thus, interindividual variation and group dynamics likely influence enrichment success, underscoring the importance of evaluating individual chimpanzee enrichment preferences as well as the social context of enrichment use [5].

Previous studies have measured enrichment use in chimpanzees, but their ethograms have been sparse in defining manipulation behaviors. For example, vague terms such as “manipulate puzzle” [21] (p. 356) or “any action directed at environment” [10] (p. 30) are employed when defining enrichment use without specifying tactile, oral, or body behaviors. Additionally, experimenters often include an abnormal behavior category in their ethograms to investigate behavioral changes but may exclude natural behaviors such as brachiation or nesting (e.g., [10]). If one of the goals of environmental enrichment is to increase natural behaviors, then ethograms should be designed to determine if enrichment objects are allowing opportunities for the expression of a range of natural behaviors.

Furthermore, while studies have assessed enrichment effects on social behaviors [11,13], the social contexts under which enrichment is used are rarely documented, and ethograms may lack sufficient detail to understand enrichment use patterns in larger social settings. For example, while neither social interactions nor aggressivity have been shown to vary in relation to the number of enrichment objects present [9], how those objects were used in larger social contexts was not examined. It is also rare for studies to categorize a wide range of device types, with studies often measuring behaviors in relation to only one type or category of enrichment (e.g., [11,13,17,23,24]). This is unfortunate given that, to fully understand how enrichment objects are used in social settings, it may be necessary to introduce and investigate a wide range of enrichment object categories.

Insight into the complex interactions among the type of enrichment object, the distinct manipulation behaviors employed by an animal, and the social contexts of enrichment use may be best provided by a large data set of potentially sporadic behaviors. Video mining represents an opportunity to study infrequent behaviors [25,26,27,28] or behaviors in private settings [29]. Crowdsourcing or video mining from video-sharing websites has been used to investigate interspecies play behavior [25], horses’ problem-solving capacities [27], humans’ responses to tail chasing in dogs [26], and the welfare of slow lorises being kept as pets [29]. Although data mining of this kind runs the risk of collecting biased reports, the approach provides a large data set of observations that reflects directly observed behavior unprompted by the researcher [28,29].

The current study evaluated three ethograms measuring distinct features of enrichment use by video mining from a large, multi-year archive of video recordings of enrichment use by captive chimpanzees in a sanctuary setting. Our ethograms expanded upon prior studies of enrichment use in captive chimpanzees by including a wider range of manipulation behaviors and object types as well as the social contexts of enrichment use. Specifically, this study evaluated if our ethograms would: (1) be sensitive enough to fully capture the range of enrichment objects, manipulation behaviors, and social contexts of enrichment use; (2) uncover patterns among specific types of enrichment objects, distinct manipulation behaviors, and different social contexts of enrichment use; and (3) allow us to characterize individual preferences in the use of objects, enrichment manipulation behaviors, and social contexts of enrichment use. By observing behavior patterns in captive chimpanzees provided with access to a wide range of enrichment objects, we also hoped to determine how to better meet the enrichment needs of those chimpanzees.

## 2. Materials and Methods

### 2.1. Study Subjects

Archival videos of chimpanzees homed at Chimpanzee Sanctuary Northwest (CSNW) in Cle Elum, WA, USA, were coded using three ethograms. Video footage included two separately housed groups of adult chimpanzees homed at CSNW between 2011 and 2021. As shown in Table 1, Group 1 consisted of six females and one male, while Group 2 consisted of two females and one male who joined the sanctuary in 2019. The primary author familiarized herself with their names and physical appearances prior to video coding. Over the years of video footage, the chimpanzees had variable access to an array of spaces, including smaller indoor rooms ranging from ~9.5 to ~13 m^2^, a large two-story indoor room, a large indoor–outdoor space with climbing structures, and a large ~1 ha open-topped outdoor space with an electric fence, earth substrate, and multiple climbing structures. Meals consisting of various fruits, vegetables, and manufactured primate chow were provided three times per day (breakfast: 10:00; lunch: 13:00; dinner: 16:30) with access to water ad libitum.

CSNW maintains an enrichment calendar specifying a daily theme for enrichment items (e.g., paper day, art day, doll day) to supplement items such as wooden and plastic toys, boots, clothing, and dolls that are provided daily. CSNW also coordinates a daily rotation of food puzzles as well as a rotation of browse three days each week during months when natural outdoor browse is limited. A large variety of enrichment objects have been provided over the years, including but not limited to hanging treat bags, drop down puzzles, forage pools, firehose cubes, blanket forts, stackable cups, shake bottles, hygiene items, children’s toys, clothing and footwear, fire hose hammocks, and a variety of technology items. Not all categories of objects are available at all times, but chimpanzees have unlimited access to a variety of objects in the indoor enclosure spaces and can bring indoor enrichment items into the outdoor spaces.

### 2.2. Ethograms

Three separate ethograms were developed by the authors by expanding and elaborating upon categories provided in prior publications (e.g., [9,10,21,30]) followed by refinement via review of videos from the CSNW video archive. Full ethograms are provided in Appendix A Appendix A. One ethogram focused on the type of enrichment object used (i.e., forage, toys, structural, nesting, technology, art, and other; see Table 2 for definition examples). Another ethogram included state and event manipulation behaviors (i.e., carry, examine, oral, play-on, vocalize, active tactile, tool, wear, nest, rest, out of view, and other; see Table 3 for definition examples;), and a third ethogram focused on the social contexts in which enrichment was used (i.e., solitary, affiliative, proximate, aggressive, and submissive; see Table 4 for definition examples). Technology objects and the examine manipulation behavior represent unique categories in the ethograms. Unlike other enrichment object categories, technology objects were held by staff while chimpanzees interacted with those objects, allowing for closer video recordings. When coding the examine behavior, it was difficult to determine if chimpanzees were looking directly at an object or simply facing toward it. In an effort to retain these categories in the ethogram to allow for evaluation, examining behavior was defined as only occurring with technology objects, and the required minimum length for the examine behavior was expanded compared to other categories in order to better confirm that the individual was looking directly at the technology object.

### 2.3. Video Mining and Data Collection

Coded videos of enrichment use ranged from 6 s to 17 min 40 s in length, were recorded by CSNW staff, and included at least one chimpanzee maintaining physical contact with an enrichment object for at least 6 s to ensure that the interaction was intentional. Examples of videos can be found at https://chimpsnw.org/enrichment-database/ (accessed on 1 February 2020). Coded videos were sampled from a total of 2054 available videos. The proportion of videos for each chimpanzee in the video archive was expected to approximately represent that chimpanzee’s relative frequency of enrichment use; therefore, we sampled videos via the following steps: (1) all videos were reviewed for occurrences of enrichment use, yielding 2539 total occurrences of enrichment use across all videos; (2) total occurrences of enrichment use by each animal per year were calculated; (3) the number of videos selected for coding for each chimpanzee was determined by the relative proportion of enrichment use by each animal per year; and (4) the specific videos for each chimpanzee for each year selected for coding were as equally distributed across that year as possible. The resultant coded sample consisted of 619 videos; four videos were not coded after redefining the behavior of examine on the ethogram. Out of the remaining 615 viable videos, 410 videos displayed only one animal using enrichment while 205 included multiple animals using enrichment. Videos with multiple animals were selected for coding based on the relative proportion of videos each year that contained multiple animals as well as the requirements of sampling individual chimpanzees. One video showing multiple individuals using enrichment was coded repeatedly and separately for each individual. In total, we coded 765 out of the 2539 total occurrences of enrichment use displayed in the video archive. The distribution of occurrences among individual chimpanzees ranged from 12 to 168 occurrences (see Appendix A).

Two individuals coded the sampled videos after multiple practice trials that were also used to refine the ethograms’ use. Interrater reliability tests on the measures in the three ethograms were, then, conducted on five videos prior to separate video coding by the two individuals. Correlations for measures from all three ethograms ranged from *r*(3)s = 0.99–1.0, *p*s < 0.001. Intrarater reliability tests were conducted on those same videos post-coding and were compared to pre-coding values, yielding similarly high correlations. Identification of chimpanzees in the videos were confirmed with filenames and, when no name was provided, were provided by an individual experienced at video identification of the chimpanzees.

### 2.4. Data Analysis

This study was largely descriptive in evaluating the efficacy of the ethograms in capturing enrichment use. Descriptive statistics included total and mean durations of interactions with enrichment object categories (i.e., forage, toys, structural, nesting, technology, art, and other), the total and mean durations for state manipulation behaviors (i.e., nest, rest, and out of view), the total and mean frequencies for event manipulation behaviors (i.e., carry, examine, oral, play-on, vocalize, active tactile, tool, wear, and other), and the total and mean durations for the social contexts of enrichment use (i.e., solitary, affiliative, proximate, submissive, and aggressive).

Consistent with previous work assessing the effectiveness of environmental enrichment [31,32], principal component analysis (PCA) was conducted to identify potential underlying patterns among the ethogram measures. The submissive social context, although in the original ethogram, was not included in PCA because no submissive contexts were observed during coding. Because variables did not meet the normality assumption for PCA, we used non-parametric Spearman’s rank-order correlations to confirm the components observed in the PCA and to further investigate potential correlations among enrichment objects, manipulation behaviors, and social contexts.

Lastly, Kruskal–Wallis tests were used to investigate individual differences in enrichment use [31,32]. Because Dwass–Steel–Critchlow–Fligner (DSCF) post hoc tests control for family-wise error [33], DSCF post hoc tests were used for significant Kruskal–Wallis tests without adjustment of alpha [34] to determine which individuals differed from each other in their use of specific objects, enrichment manipulation behaviors, and social contexts of enrichment use. Jamovi 1.6.23 software was used for all statistical analyses.

## 3. Results

In general, the three ethograms were sensitive enough to fully capture the range of enrichment objects, manipulation behaviors, and social contexts of enrichment use observed in the sampled videos. As noted, the sample did not contain any observations of submissive social contexts of enrichment use. The majority of enrichment use was accounted for via our ethograms; however, several objects and behaviors were coded as other. Specifically, items such as magazines, books, bags, and hanging mirrors were coded as other in the enrichment object ethogram and accounted for 447 s of the total duration of all coded objects. There were 18 total instances of manipulation behaviors that could not be coded with established behavioral categories, including spitting out objects, balancing on a tightrope, and sniffing objects and were, therefore, coded under other in that ethogram.

### 3.1. Descriptive Statistics

Table 5 displays the percentage of sampled videos in which an ethogram category was observed as well as the total and mean duration or frequency per occurrence for specific enrichment object use and manipulation behaviors. As presented in Table 5, toy and forage object use occurred often in the sampled videos and had relatively long total durations and mean durations of use per occurrence. Similarly, oral and active tactile manipulations occurred very often in the sampled videos with high frequencies across the sample and per occurrence. Total durations were also calculated for social contexts of enrichment use, including 248 min and 49 s for solitary contexts (*M* = 19.5 ± 30.7 s), 160 min and 22 s for proximate contexts (*M* = 12.6 ± 29.0 s), 139 min and 21 s for affiliative contexts (*M* = 10.9 ± 28.4 s), and 32 s for aggressive contexts (*M* = 0.0 ± 1.1 s). Overall, toy and forage object use and solitary contexts of enrichment use as well as oral and active tactile behaviors were very common in the sampled coded videos.

### 3.2. Principal Component Analysis (PCA)

Shapiro–Wilk tests indicated significant departures from normality (*p*s < 0.001) for all measures, including durations of use of enrichment objects, frequencies and durations of manipulation behaviors, and durations of social contexts of enrichment use. Per PCA assumptions of linearity, we removed five outlier occurrences of enrichment use that contributed to curvilinear relationships among the variables (i.e., two outliers involving use of forage objects, one outlier each involving a structural and nesting object, and one outlier of resting behavior). Visual inspections of scatterplots of the remaining 760 occurrences of enrichment use revealed non-curvilinear relationships among all variables. Consistent with other investigations of enrichment use [31,32], PCA was conducted to identify potential underlying components among the 23 coded variables, including the duration of use of specific categories of enrichment devices (i.e., forage, toys, structural, nesting, technology, art, and other), the frequency of manipulation behaviors (i.e., carry, examine, oral, play-on, vocalize, active tactile, tool, wear, and other), the duration of manipulation behaviors (i.e., nest, rest, and out of view), and the duration of enrichment use in social contexts (i.e., solitary, affiliative, proximate, and aggressive).

PCA using varimax rotation with the loading threshold set to 0.60 yielded five components with eigenvalues greater than 1.4. The first component accounted for 9.6% of the total variance in the variables, the second component for 9.6%, the third component for 8.3%, the fourth component 7.4%, and the fifth component for 7.3%. Each component had positive loadings and included two to three variables. As shown in Table 6, Components 1 and 2 (i.e., Social Toy Manipulation and Solitary Foraging Manipulation, respectively) were both labeled in order to highlight the inclusion of a social context variable along with an enrichment object and manipulation behavior variable. As expected given the ethogram category labels, components 3, 4, and 5 confirmed that specific objects and behaviors co-occurred, including technology objects with examine behavior (loadings ≥ 0.91), nesting objects with nesting behavior (loadings ≥ 0.65), and structural objects with play-on behavior (loadings ≥ 0.78). The remaining ten variables did not load onto any component.

### 3.3. Spearman’s Rank-Order Correlations

As previously noted, linear relationships were confirmed via visual inspections of scatterplots. Because variables displayed non-normality, Spearman’s rank-order correlations were utilized to confirm the patterns among variables identified in the PCA and investigate potential additional relationships. Because the primary objective in evaluating our ethograms was focused on identifying potential relationships across ethograms, only cross ethogram category associations were investigated. As indicated in Table 7, Spearman’s rank-order correlations confirmed the relationships among variables in all five components identified in the PCA.

#### Additional Relationships

As detailed in Table 7, Spearman’s rank-order correlations revealed 17 additional relationships among enrichment objects, manipulation behaviors, and social contexts that were not represented in the PCA. Overall, toy objects showed more positive correlations with distinct manipulation behaviors than did other enrichment objects. Forage objects were associated with decreased affiliative but increased proximate social contexts and, likely because forage object use was strongly associated with oral manipulation behaviors, forage object use was negatively correlated with active tactile, wear, and resting manipulation behaviors. Art objects were only correlated with tool use and had no association with social contexts, while other objects were only correlated with solitary contexts. Three behaviors (i.e., vocalize, other, and out of view) and one social context (i.e., aggressive) did not correlate with any of the enrichment objects and are not included in Table 7.

### 3.4. Kruskal-Wallis Tests

In order to evaluate if the ethograms could be used to investigate individual differences in enrichment use, Kruskal–Wallis tests were conducted for each of the 23 measured variables. The omnibus Kruskal–Wallis tests indicated that the 10 chimpanzees significantly differed in 10 out of 23 categories, including the duration of the use of several enrichment object categories (*χ*^2^(*9*)s = 33.5–150.6, *p*s < 0.001, ε^2^s = 0.04–0.20) as well as the frequency of some manipulation behaviors (*χ*^2^(*9*)s = 32.0–65.6, *p*s < 0.001, ε^2^s = 0.04–0.09) and duration of one social context (*χ*^2^(*9*) = 31.5, *p* < 0.001, ε^2^ = 0.04). Follow-up DSCF post-hoc tests [33,34] confirmed individual differences in only six categories, including four objects (i.e., forage, toy, nesting, and other objects) as well as two manipulation behaviors (i.e., play-on and active tactile behaviors) as shown in Table 8.

Overall, as displayed in Table 8, toy and forage object use showed significant differences between multiple individuals. Burrito and Foxie had more toy use than many other chimpanzees, and Negra engaged more with forage objects than did Burrito, Foxie, and Honey B. With regard to manipulation behaviors, the frequency of active tactile behavior showed the most differences between individuals with Negra showing less active tactile behavior than Burrito, Foxie, Jamie, and Honey B. The frequency of play-on behavior also differed with Missy showing more frequent play-on behavior than Burrito and Jamie.

## 4. Discussion

In the current study, chimpanzees spent a lot of time engaged with toy and forage objects as compared to structural, nesting, technology, art, and other objects. When the chimpanzees manipulated objects, they commonly used oral and active tactile behaviors. Importantly, while the chimpanzees spent considerable time using enrichment objects under solitary contexts and forage object use occurred with oral behaviors primarily in solitary contexts, toys stimulated active tactile behavior in affiliative contexts. Forage objects were also positively associated with proximate contexts but negatively associated with affiliative contexts. Both forage objects and toys were correlated with a larger variety of manipulation behaviors than were other types of objects. Individual animals also had clear preferences for objects. For example, Negra, a wild-born chimpanzee, preferred to use forage objects while Burrito and Foxie preferred to use toys. Overall, the ethograms were mostly effective in capturing enrichment use and revealing patterns of object use, behavior, and social context in chimpanzees with the exception of a few objects and manipulation behaviors.

### 4.1. Ethogram Efficacy

In general, we found that our ethograms were effective in capturing the range of enrichment objects, manipulation behaviors, and social contexts observed in our sample. Enrichment studies tend to investigate one form or category of enrichment [11,13,17,23,24] rather than multiple enrichment categories simultaneously [12]. The current findings indicate that captive chimpanzees will voluntarily interact with a wide variety of objects when available. Although all enrichment object use was accounted for with our ethograms, objects such as mirrors, books, and magazines were difficult to place into established ethogram categories and were, therefore, coded under the “other” object category. Many of the objects in the “other” category seemed to provide visual stimulation to chimpanzees, suggesting that visually engaging objects may be particularly appealing.

Details regarding specific tactile, oral, or body behaviors used during enrichment have also been lacking in the literature [10,21] despite the call to provide sensory enrichment in addition to physical, social, food, and cognitive or occupational enrichment [4]. Our findings reveal that chimpanzees use a vast array of distinct behaviors when interacting with enrichment objects. Some manipulation behaviors were difficult to code, including spitting out objects and sniffing objects. Sniffing may be particularly important to include in future ethograms because it is an information-gathering behavior for chimpanzees [35]. We especially encountered difficulties with the examine behavior, indicating that eye tracking studies may be needed to measure engagement with some objects perhaps because chimpanzees shift their fixation location more quickly and broadly than do humans [36]. However, our inability to capture examine behaviors with other types of objects did not undermine our ability to capture other behaviors and, while some objects co-occurred with certain behaviors (e.g., nesting objects with nesting behavior), ethogram labels of objects did not necessarily influence the behaviors that were captured with those objects.

Lastly, while prior studies have assessed the effects of enrichment on social behaviors [11,13] or evaluated the role of rank [20] or housing conditions [16,19] on enrichment use, how enrichment objects are used in larger social contexts remains unexamined. In the current study, we found solitary, proximate, and affiliative contexts of enrichment use to be common with solitary contexts resulting in longer use of objects. No submissive contexts and very low levels of aggressive behavior were observed. Submissive behaviors might have been too brief to capture or may not have been employed by the two captive groups in our sample, and low levels of aggression may have occurred because the chimpanzees in the current study did not have to compete for access to enrichment objects [17,21].

### 4.2. Group Preferences for Objects

Out of the seven enrichment object categories included in our ethogram, the chimpanzees in the sample spent considerable amounts of time engaged with toys and forage objects. Although naturalistic enrichment might be more preferred by caregivers and the public due to its visual appeal and similarity to the natural environments of chimpanzees, the current finding that several chimpanzees in our sample preferred artificial and “unnatural” toy objects over all other types of objects suggests that only providing naturalistic enrichment may be limiting. For example, one of the chimpanzees, Foxie, is uninterested in most types of enrichment aside from troll dolls (J.B. Mulcahy, personal communication, 30 March 2022); a preference that would have remained unknown had the facility limited enrichment to only naturalistic items. Furthermore, authors have argued that the efficacy of an object in improving animal welfare should have greater weight than the naturalness or unnaturalness of the object when being considered for enrichment implementation [37]. Because previous studies have suggested that toys as enrichment may improve chimpanzee welfare as indicated by a decrease in abnormal behaviors [8,9], unnaturalistic objects should be considered for inclusion in enrichment items.

### 4.3. Objects, Manipulation Behaviors, and Contexts

In evaluating our ethograms, we were particularly interested in whether patterns among enrichment objects, specific manipulation behaviors, and social contexts would emerge. Our finding of solitary foraging and social toy manipulation underscores the importance of multifaceted measures of enrichment use. While the relationship between forage objects and oral behavior is to be expected, forage objects were also associated with both solitary and proximate contexts. These findings mirror feeding in wild chimpanzees, which is largely a group activity with some individuals moving away from the group to feed alone [38]. Although forage objects may have resulted in proximate contexts in captive chimpanzees because of the objects’ placement, the solitary foraging manipulation observed in the current study indicates that caregivers of captive chimpanzees should ensure that individuals have enough space to spread out when forage objects are presented.

Social toy manipulation consisted of toy use, active tactile manipulation, and affiliative contexts, which may represent object play. This pattern is consistent with observations that small objects promoted fine motor engagement and object play with other individuals in wild chimpanzees [39,40,41]. However, object play may serve different functions in wild compared to captive chimpanzees. For example, engagement in object play was rarely observed among wild adult chimpanzees compared to infants, juveniles, and adolescents [41]; a finding which contrasts with the current sample of adult captive chimpanzees who engaged in toy play. Such differences might be due to wild chimpanzees needing to spend a larger portion of their activity budgets engaged in survival behaviors such as predator avoidance [38] and food searching [42]. Therefore, caregivers of captive chimpanzees should expect that small, manipulatable, non-food objects will provide opportunities for object play while keeping individuals socially engaged.

Additional relationships among enrichment object categories, manipulation behaviors, and social contexts were also identified independently of the social toy and solitary foraging patterns. Both toy and art objects were associated with tool use, suggesting that these objects allowed opportunities for species-typical tool use behaviors that have been observed in the wild [38,43]. Like toys, nesting objects were associated with affiliative use; mirroring the finding that captive chimpanzees often engage in affiliative behavior and maintain physical contact with others while nesting [44]. Overall, our ethograms captured engagement in many different species-typical behaviors and social activities, with toys associated with the widest variety of manipulation behaviors, and forage objects associated with increased solitary and proximate contexts but decreased affiliative contexts.

Neither vocalizations nor aggressive social contexts were associated with other ethogram categories in the current study. Chimpanzees produce a wide range of unique vocal sequences [45] and the current finding suggests that chimpanzees are not selectively vocal in response to particular categories of enrichment objects nor in conjunction with enrichment use in specific social contexts. With regard to aggressive social contexts, the chimpanzees in the current study had access to a wide variety and number of enrichment objects which they could use throughout the facility, removing the need to compete for a limited resource [21]. For objects with limited access (i.e., technology objects), staff members strictly controlled contact, which may have mitigated potential aggression in response to having only one device available.

### 4.4. Individual Differences

When subjects are free to choose whether to engage in enrichment activities, participation is usually assumed to be a positive indicator of interest or motivation [46]. Therefore, we also evaluated our ethograms for their ability to reveal individual differences in enrichment use, behaviors, or social contexts. We found that individuals differed in their use of enrichment objects and manipulation behaviors, but not in the social contexts of enrichment use. For example, Negra spent less time using toys but more time using forage objects when compared to other chimpanzees, indicating that Negra may be particularly food-motivated. The chimpanzees in the current sample also showed individual differences in some manipulation behaviors, differing in the frequencies of their active tactile and play-on behaviors. Perhaps because of her comparative lack of interest in toys, Negra less often engaged in active tactile behaviors, whereas Missy had a greater frequency of play-on behavior than did other chimpanzees.

Individual differences in object use and behaviors support the proposition that age, sex, past experiences, and temperament influence an animal’s enrichment needs and should, therefore, be taken into consideration by caregivers [15,37,47]. Negra was the only chimpanzee in our sample confirmed to be wild-born, and her activity patterns are similar to those of wild chimpanzees who spend more time foraging than do captive chimpanzees [42] and manipulate objects less often at older ages [40,41]. In contrast, Burrito and Foxie, both captive-born, spent considerable time with toys. While Burrito’s toy use appears inconsistent with at least one prior report of less toy use by adult captive male chimpanzees [21], object manipulation is higher in wild juvenile males than females [48], suggesting that captivity may alter species-typical developmental trajectories. Foxie‘s focus on troll doll toy use may be related to a pre-sanctuary history that included removal of multiple offspring from her care [49]. Thus, for Foxie, troll dolls may be similar to an infant in soliciting species-typical mothering behavior [50]. Lastly, Missy had more play-on behaviors, supporting her nickname among sanctuary staff as the “athlete” [51,52,53]. These findings, in conjunction with a growing body of evidence demonstrating personality differences in non-human primates (see [54] for a review), suggest that having a wide variety of enrichment objects may be an essential component to creating individually fulfilling captive environments.

### 4.5. Video Mining and Limits to Generalization

Our study sampled from a larger video library, and observer bias may have occurred during video collection because staff members may have preferred to record interactions involving specific categories of enrichment objects. Video mining does share similar limitations to the selection bias of human subjects volunteering or to field ethology studies that use animals that happen to present themselves to the observer or trail camera [28]. In our sample, potential observer biases or effects may have been ameliorated, at least in part, by the fact that videos were collected by multiple staff members, potentially as many as 15 to 20 individuals, over a ten-year period. As with other analyses of video-sharing websites or studies with small samples, the current findings require caution in generalizing conclusions beyond the sample population. For example, because our sample consisted of only two males and eight females, conclusions regarding the impact of sex on enrichment use were not possible. However, consistent with the documented benefits of video mining [25,28,29], our sample from a large, multi-year private video archive provides valuable insight into the nuanced interactions among enrichment objects, distinct manipulation behaviors employed by an animal, and the social contexts of enrichment use in chimpanzees in a sanctuary setting.

## 5. Conclusions

In the current study, chimpanzees as a group spent considerable time with toy and forage enrichment objects. However, the extent of those preferences largely depended on the individual, indicating that enrichment provisions should account for individual needs. Additionally, some behaviors, objects, and contexts clustered together, revealing complex behavior patterns such as solitary foraging and social toy manipulation. As well, chimpanzees in the current study mainly engaged in solitary enrichment use, with the social contexts of enrichment use largely dependent on the types of objects being used, indicating that enrichment objects can play different roles in social relationships. Despite limitations, this study indicates that more nuanced and detailed ethograms accounting for enrichment use can be effective in capturing object preferences, engagement in manipulation behaviors, and social contexts of enrichment use. It is hoped that these findings will offer potentially useful ideas for caregivers at other facilities as well as allow caregivers to better meet the behavioral and cognitive needs of animals in captivity. Lastly, we hope that this evaluation of detailed ethograms designed to capture distinct elements of enrichment use in chimpanzees fosters future research that evaluates how and in what contexts chimpanzees use specific categories of enrichment.

## Figures and Tables

**Table 1 animals-12-02029-t001:** Demographics of each participant during the study period.

	Name	Sex	Approx. Age Range	Birth Setting
Group 1				
	Annie	F	36–47	Unknown
	Burrito	M	27–38	Captivity
	Foxie	F	34–45	Captivity
	Jamie	F	33–44	Captivity
	Jody	F	35–46	Unknown
	Missy	F	35–46	Captivity
	Negra	F	37–48	Wild
Group 2				
	Honey B	F	28–32	Captivity
	Mave	F	28–32	Captivity
	Willy B	M	27–31	Captivity

**Table 2 animals-12-02029-t002:** Enrichment object ethogram.

Object	Definition
Forage	Food or objects in other categories ^a^ that are linked to obtaining food. May include produce, bamboo, flowers, toys containing food, plates with food, cups with drink, chow (only when obtained from object), pools with food or water, bags with food, snow (only when indoors for enrichment purposes), object containing snow (e.g., bucket), hay bags, or straw containing food.
Toy	Smaller objects that allow utilization of fine motor skills and do not contain food or drink. May include balls, clothing, paper braids, pipes, puzzle devices, hammers and other tools, cleaning utensils, sandbox with sand, dolls, soap bubbles, plates, cups, straws, or paper bags.
Structural ^b^	Enclosure structures and larger objects that allow perching, sitting, or utilization of gross motor skills such as climbing. May include barrels, pools, scooters, tires, cardboard boxes, ropes, bridges, firehoses, ladders, artificial trees, stools, tables, tunnels, or platforms.
Nesting	Flexible objects that can be flattened or otherwise manipulated to form a nest. May include blankets, hammocks, cardboard paper, butcher paper, or straw.
Technology	Digital devices, with devices remaining visible. May include iPad, digital camera, cell phone, or GoPro.

^a^ When an individual obtained food from within or on top of an enrichment object, both contact with object and contact with food were coded under the same duration. ^b^ Objects coded as structures had to be optional and avoidable. For example, a platform that was visibly part of the path from the indoors to the outdoors would have been unavoidable and, therefore, did not count as a structure.

**Table 3 animals-12-02029-t003:** Enrichment manipulation behavior ethogram.

Behavior	Definition
Oral ^a^	Individual brings food or nonfood enrichment object to its mouth and may bite, chew, kiss, suck, or lick object.
Active tactile ^a,b^	Individual moves or manipulates object with body part, either with object moving or body part moving against object.
Play-on ^a^	Individual brachiates, swings, hangs, or climbs on enrichment.
Nest ^c^	Individual surrounds themselves with objects, pulls objects in close, or flattens or otherwise manipulates objects to form a nest around their body.
Examine ^a^	Individual’s gaze is focused on or head is facing towards digital technology for at least 10 s without breaking gaze, moving head away from, or manipulating object.

^a^ Manipulation behavior was defined as an event behavior and measured in frequency. ^b^ Did not include nest, tool, rest, or wear behaviors; bringing object to mouth; removing from mouth; or touching object while not moving object or body. ^c^ Manipulation behavior was defined as a state behavior with the duration of state behaviors measured in seconds.

**Table 4 animals-12-02029-t004:** Social context of enrichment use ethogram.

Context	Definition
Solitary Use	Individual uses object alone without interacting with other chimpanzees, with no other chimpanzees interacting with object, without being in proximity (less than two arms’ length) to other chimpanzees, and without engaging in affiliative use with staff.
Affiliative Use ^a^	Individual uses object while touching staff, uses object with staff touching object, gives object to staff, or plays with staff while touching object. Also includes playing or interacting with other chimpanzees while touching object or using object together. Individuals may exhibit play, grooming, or laughing while interacting.
Proximate Use	Individual uses object with other chimpanzees in proximity (less than two arms’ length) and without other chimpanzees interacting with the object or individual.

^a^ Did not include staff giving food to individual.

**Table 5 animals-12-02029-t005:** Percentage (%) of sampled videos in which the ethogram category was present and total and mean (±SD) durations and frequencies for object use and manipulation behaviors (*n* = 765 occurrences; 615 videos; 732 min, 58 s of video).

	% of Videos	Total Duration in Sample	Mean Duration (s) per Occurrence		% of Videos	Total Frequency in Sample	Mean Frequency per Occurrence
Objects				Behaviors			
Toy	41.4%	181 min, 25 s	14.2 ± 28.6	Oral	62.6%	1786	2.3 ± 3.6
Forage	41.0%	163 min, 0 s	12.8 ± 34.4	Active tactile	52.0%	1275	1.7 ± 2.9
Structural	25.9%	102 min, 30 s	8.0 ± 19.2	Tool	9.9%	358	0.5 ± 2.4
Nesting	2.9%	73 min, 29 s	5.8 ± 18.4	Carry	17.9%	258	0.3 ± 1.0
Other	2.9%	7 min, 27 s	0.6 ± 4.2	Vocalize	7.0%	96	0.1 ± 0.6
Technology	0.8%	5 min, 32 s	0.4 ± 4.6	Play-on	7.6%	93	0.1 ± 0.5
Art	0.7%	2 min, 57 s	0.2 ± 3.8	Wear	7.3%	77	0.1 ± 0.5
Behaviors				Examine	0.8%	13	0.02 ± 0.2
Out of view	48.5%	268 min, 54 s	21.1 ± 40.2	Other	2.4%	18	0.02 ± 0.2
Nesting	3.1%	7 min, 32 s	0.6 ± 5.5				
Resting	4.1%	6 min, 45 s	0.5 ± 4.9				

**Table 6 animals-12-02029-t006:** Component loadings for enrichment objects, manipulation behaviors, and social contexts (*n* = 760 occurrences of enrichment use).

Component	Variables	Loading
1: Social Toy Manipulation		
	Toy object	0.85
	Active tactile behavior	0.65
	Affiliative context	0.74
2: Solitary Foraging Manipulation		
	Forage object	0.81
	Oral behavior	0.80
	Solitary context	0.60

**Table 7 animals-12-02029-t007:** Spearman’s rank correlation coefficient (Spearman’s rho) for objects, behaviors, and social contexts (*n* = 760 occurrences of enrichment use).

					Object			
		Toy	Forage	Nesting	Structural	Technology	Art	Other
Behavior								
	Oral	−0.061	0.638 *^a^	−0.189 *	−0.283 *	−0.089	0.046	−0.089
	Tool use	0.153 *	0.056	−0.061	−0.043	−0.030	0.174 *	−0.016
	Active tactile	0.411 *^a^	−0.190 *	0.100	−0.102 *	−0.076	0.072	0.033
	Wear	0.199 *	−0.169 *	0.193 *	−0.077	−0.026	−0.018	0.040
	Examine	−0.042	−0.080	0.030	−0.062	1.000 *^a^	−0.008	−0.016
	Play-on	−0.106	−0.080	−0.077	0.375 *^a^	−0.029	−0.021	−0.044
	Nesting	−0.054	−0.107	0.397 *^a^	−0.081	−0.017	−0.012	0.030
	Resting	−0.006	−0.122 *	0.228 *	−0.025	0.057	−0.013	0.022
	Carry	0.254 *	0.012	−0.104	−0.053	−0.045	−0.032	−0.046
Social Context								
	Solitary	0.001	0.145 *^a^	−0.028	0.064	−0.088	0.089	0.199 *
	Affiliative	0.334 *^a^	−0.353 *	0.263 *	−0.056	−0.002	−0.041	−0.024
	Proximate	−0.051	0.140 *	−0.063	0.125 *	0.113	−0.059	−0.078

Note. Only variables with significant correlations are included in the table. See text for details. * *p* < 0.001; ^a^ Correlation confirming PCA.

**Table 8 animals-12-02029-t008:** Mean duration of enrichment object use and mean frequency of manipulation behaviors for each individual (SDs in parentheses; *n* = 760).

Individual	Objects (In Duration)				Behaviors (In Frequency)	
	Toys	Forage	Nesting	Other	Active Tactile	Play-On
Annie	5.7 ^ab^ (18.2)	12.1 (22.4)	8.3 (18.4)	0.0 (0.0)	0.8 (1.7)	0.2 (0.6)
Burrito	24.5 ^acde^ (40.4)	10.9 ^a^ (23.8)	2.0 (9.43)	0.5 (3.9)	2.3 ^a^ (3.4)	0.1 ^a^ (0.4)
Foxie	24.2 ^bfghi^ (29.7)	5.2 ^bc^ (11.2)	2.6 ^a^ (12.5)	0.0 ^a^ (0.0)	2.1 ^b^ (3.7)	0.1 (0.5)
Jamie	18.5 ^jkl^ (31.9)	12.8 (25.7)	6.5 (20.3)	1.4 (6.2)	2.0 ^c^ (2.7)	0.1 ^b^ (0.4)
Jody	2.4 ^cfjm^ (9.7)	15.4 ^b^ (20.4)	12.2 ^a^ (23.9)	0.0 (0.0)	1.1 (2.4)	0.1 (0.6)
Missy	2.4 ^dgkn^ (8.4)	14.0 (29.5)	5.2 (15.6)	0.7 (6.1)	1.1 (2.0)	0.4 ^ab^ (0.8)
Negra	6.4 ^ehl^ (23.8)	19.5 ^acd^ (25.1)	3.0 (10.2)	0.0 (0.0)	0.4 ^abcd^ (1.1)	0.0 (0.1)
Honey B	17.1 ^mn^ (25.7)	6.1 ^d^ (22.8)	8.0 (21.4)	1.6 (7.1)	3.1 ^d^ (3.3)	0.1 (0.5)
Mave	3.1 (6.5)	4.6 (8.8)	1.3 (3.2)	3.1 ^a^ (7.5)	0.6 (0.9)	0.0 (0.0)
Willy B	2.0 ^i^ (6.5)	7.1 (10.5)	4.2 (12.1)	0.0 (0.0)	0.9 (1.5)	0.0 (0.0)

Note. Superscripts indicate significantly differences between individuals (*p* < 0.001).

## Data Availability

The data presented in this study are available upon reasonable request from the corresponding author.

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
