# Peer review of "An Evaluation of Ethograms Measuring Distinct Features of Enrichment Use by Captive Chimpanzees (Pan troglodytes)"

_animals, 2022, doi:10.3390/ani12162029_

Round 1

Reviewer 2 Report

Overall this paper on ethogram suitability for enrichment use in captive chimpanzees is a reasonably useful and well written piece of work that will add to the bulk of research available. This is a heavily explored topic but specific fine focused ethograms for detailed enrichment studies has been less well explored and so I agree that there is a use for this, though more could have been made in the discussion and indeed in the title to make this stand out as a unique piece of work.

The introduction sets up the topic but is a little on the short side and misses some important work such as that by Buchanan-Smith, as well as an overview or at least a link to research on enrichment evaluation, which highlights the need to have such fine scale enrichment behavioural studies.

Abstract line 31 -33 long sentence needs reworded for clarity and broken up

Section 2.3 please state what software/packages were used for the analysis.

A large number of Kruskal Wallis tests were carried out with post hoc test, what significance value was used and what steps were taken to minimise type 1 errors. Could these be more refined with specific focus and justification?

Suggest some graphs could be included to help visualise the key findings.

Were any agonistic behaviours explored in the ethogram - submissive behaviours were mentioned but what about other agonistic behaviours?

The discussion brings in the key points and refers to wider literature but more depth could have been brought in for example- what explanation is there for Burrito and Foxie having more toy use and lower forage object use? Could the proximity to forage items be related to where they were placed rather than how the animals have chosen to congregate?

Please relate the findings more to the literature in the intro to tie the work together better.  

Reviewer 3 Report

The study presents an investigation of the enrichment of chimpanzees and the associations between specific enrichment types, behaviour types and social contexts. The study attempts to fill a gap on our understanding of environmental enrichment in chimpanzees but lacks the scientific rigor and validity to do so.

There are some serious methodological problems with the study. Many of the problems are the result of poorly written descriptions of what was done but some are more fundamental. Firstly, very little environmental context and procedural information is provided. How an individual animal chooses to interact with their environment (including enrichment) is determined by what the nature of the environment is. This in turn determines how the observed interactions with the environment can be interpreted in a broader sense. Without this information it is very difficult for the reader to interpret the rest of the described methodology and the discussion, particularly as it relates to the definitions of behaviours and contexts.

The definitions of the behavioural contexts under consideration are inappropriate, being either unnecessarily restrictive or excluding aspects which are seemingly obvious. For example, ‘examine’ behaviour can only be applied to electronic devices; this precludes any behaviour where the animal obtains sensory information from a non-electronic object. A chimpanzee feeling the texture of cardboard or a blanket thus cannot be considered as the animal examining the object. The definition used for tool use does not mirror what is commonly considered tool use and lacks the nuance of the concept. Another example is the enrichment types under consideration. The authors classify certain objects as being ‘structural’ or ‘nesting’ objects but their uses are not restricted to these classifications. Instead of observing the chimpanzees’ interactions with the objects and classifying the behaviour, the authors are prescriptively assigning a function to the object in a manner which is misleading in how it is interpreted (e.g. labelling an object as ‘nesting’ implies it is used for nesting whereas this may not be the case). The definitions underpin the science of what has been done and how it can be interpreted; if these are not clear or are illogical, this undermines the entire study.

The data collection procedure described here is vague and contradictory. Given the provided description, it would be nearly impossible to replicate this study. Even basic methodological elements, such as how behaviour was recorded and the recording rules used, are not clearly described. It is not clear why certain methodological choices were made and the authors do not try to justify them (for example, some measures were recorded as durations whereas others were recorded as frequencies and the reason for this is never addressed).

There is also the problem that the methodology described does not represent a sample of the behaviour of the chimpanzees in the true sense. The videos used were captured [presumably at random, but no clear context for the video capture is provided] over several years and are used to assess individual differences in enrichment use. The videos do not sample all individuals in all contexts nor are the videos necessarily captured in a systematic way and thus cannot provide a representative sample of the enrichment use patterns of the individuals under consideration. Thus, assessing individual differences is not possible without recognising that these differences may be biased on the basis of a limited number of records. There is also the unjustified sub-sampling to consider which further complicates any interpretation of the study.

The statistical design has major problems. The described methods do not appear to match the results which are presented and are severely lacking in details which are critical for the reader to assess their appropriateness. Based on the information provided, the authors appear to be using tests with data which violate the statistical assumptions of those tests. The authors do not describe the structure of the models and tests used in the study and thus the reader cannot assess how appropriate the tests are in relation to the data and the study aims.

The problems with the methodology translate into poor interpretation which erodes at the value of the science. For example, the first hypothesis relating to the association between specific objects and certain contexts seems to have been engineered into a self-fulfilling hypothesis, when considering some of the methodology as presented. The three components which cross-validated the ethogram (associations between ‘examination’ behaviour and electronic technology, ‘nesting’ behaviour and nesting objects and ‘climb-on’ behaviour and structural objects) were constructed such that they could not fail to cross-validate when tested: in all three cases the behaviour in question is specifically defined as involving the types of objects highlighted in the associations and thus precludes the involvement of any other object type. Thus the logic applied by the authors through this analysis is circular and thereby self-fulfilling.

There is also a tendency of the authors to consider the findings apparently without giving thought to the biological context of the study. The social dynamics of chimpanzees in captivity are known to have a great influence on the behaviour of the individuals and this does not seem to be accounted for in either the methodology or the discussion. There is a general sense that the authors are not familiar with the biology of their study species as well, given the many ways in which the authors apply concepts in a rigid and restrictive manner to the behaviour they are reporting on.

Given the above there is little that can be done to salvage this study for publication. It is therefore my recommendation that this study be rejected.

Round 2

Reviewer 1 Report

The MS is much improved post-editing, congratulations! I particularly appreciate that the central aim of the work is now much clearer. This not only strengthens the MS as a whole, but also places some of the other concerns I had in context. I also appreciate the work you have done to expand the methods sections; this also greatly improves the MS readability, overall coherence, and potential reproducibility. I'm glad because it's interesting work.

Reviewer 3 Report

The authors have made a concerted effort to improve the manuscript. The manuscript reads well now and the authors have clarified the writing such that the study is repeatable and understandable to someone not directly involved therein. There is also a meaningful alignment between the study aims, methods and results, which was not evident previously. I am recommending that the manuscript be accepted for publication.